# Rural Women Have a Prolonged Recovery Process after Esophagectomy

**DOI:** 10.3390/cancers16061078

**Published:** 2024-03-07

**Authors:** Julia Schroeder, Kiran Lagisetty, William Lynch, Jules Lin, Andrew C. Chang, Rishindra M. Reddy

**Affiliations:** 1University of Michigan Medical School, 3808 Medical Science Bldg, Ann Arbor, MI 48109, USA; 2Michigan Medicine, Section of Thoracic Surgery, Department of Surgery, 1500 E. Medical Center Drive, TC 2120, Ann Arbor, MI 48109, USA

**Keywords:** esophageal cancer, esophagectomy, peri-operative outcomes and complications, health disparities, gender disparity, geographic access to care

## Abstract

**Simple Summary:**

Geographic access to care plays a large role in health disparities, especially with respect to esophageal cancer care. Given the increased regionalization of complex surgeries to major treatment centers, individuals from rural areas are required to travel further distances to access these complex healthcare services. Previous research has also shown gender disparities in surgical outcomes, with women less likely to receive curative cancer surgery. The aim of our retrospective study was to assess the peri-operative outcomes in patients undergoing esophagectomy for loco-regional esophageal cancer at a single academic tertiary care center based on geographic home location and gender. We found that rural women had a more complex inpatient recovery process (a longer length of hospital stay and a higher likelihood of being admitted to ICU) after esophagectomy compared with female metropolitan or male counterparts. Future studies focusing on the impact of pre-operative and post-operative interventions are needed to understand and eliminate disparities for rural-based females.

**Abstract:**

Background: Gender and geographic access to care play a large role in health disparities in esophageal cancer care. The aim of our study was to evaluate disparities in peri-operative outcomes for patients undergoing esophagectomy based on gender and geographic location. Methods: A retrospective cohort of prospectively collected data from patients who underwent esophagectomy from 2003 to 2022 was identified and analyzed based on gender and county, which were aggregated into existing state-level “metropolitan” versus “rural” designations. The demographics, pre-operative treatment, surgical complications, post-operative outcomes, and length of stay (LOS) of each group were analyzed using chi-squared, paired *t*-tests and single-factor ANOVA. Results: Of the 1545 patients, men (83.6%) and women (16.4%) experienced similar rates of post-operative complications, but women experienced significantly longer hospital (*p* = 0.002) and ICU (*p* = 0.03) LOSs as compared with their male counterparts, with no differences in 30-day mortality. When separated by geographic criteria, rural women were further outliers, with significantly longer hospital LOSs (*p* < 0.001) and higher rates of ICU admission (*p* < 0.001). Conclusions: Rural female patients undergoing esophagectomy were more likely to have a longer inpatient recovery process compared with their female metropolitan or male counterparts, suggesting a need for more targeted interventions in this population.

## 1. Introduction

Esophageal cancer remains a prevalent cancer worldwide, currently ranking as the eighth most common type of cancer and the sixth most common cause of cancer-related deaths with an expectation that the incidence will continue to rise over the next decade [1,2,3]. The incidence of esophageal cancer is higher in Asia and Africa, with nearly four out of five cases occurring in non-industrialized nations [1,4]. Yet esophageal cancer in the Western world, especially in the United States is still significant. In fact. the American Cancer Society estimates that, in the United States, there will be approximately 22,370 new esophageal cancer cases diagnosed (17,690 in men and 4680 in women) and about 16,130 deaths (12,880 in men and 3250 in women) from esophageal cancer in 2024 alone [4]. 

Globally, esophageal squamous cell cancer remains the predominant histological subtype, but interestingly, the epidemiology of esophageal cancer in the Western world has changed significantly [1]. In North America, Western Europe, and Australia, esophageal adenocarcinoma is now the most common histological subtype, and there has been a decline in the incidence of squamous cell cancer [1,4,5]. It is believed that this significant increase in adenocarcinoma in the Western world coincides with an increase in rates of gastroesophageal reflux disorder (GERD) and obesity [1]. Other known risk factors for esophageal cancer include but are not limited to age, sex, tobacco, alcohol, diet, insults to the esophageal lining (i.e., chemicals), and infections such as Human Papilloma Virus and *H. Pylori* [1,4]. 

Compared with other cancers with high prevalence rates, esophageal cancer continues to carry a poor prognosis, with an overall five-year survival between 15 and 20% [1]. These persistent poor outcomes have resulted in complex and multidisciplinary approaches to care with numerous visits to multiple specialists. For mucosal-based tumors that are limited to the mucosa (Stage 0) or invade the lamina propria without lymph node or distal involvement (Stage TIa), the mainstay of treatment is endoscopic resection. Resections for Stage I tumors have a success rate of 91 to 98% [5]. For Stage TIb tumors (extending through the muscularis mucosa into the submucosa), a lymphadenectomy is recommended given the high risk of lymph node spread [5]. For patients with potentially curable localized tumors (Stages IIA/IIB), surgical resection via esophagectomy is the primary treatment technique. More advanced regional disease (Stage III) requires a multimodal aggressive approach with neoadjuvant chemotherapy and radiation [1,2,5]. For patients with residual or recurrent disease after complete resection, typically, adjuvant chemotherapy or radiation is used, but there is no good evidence at this time [5]. For individuals with Stage IV esophageal cancer or nonresectable tumors, palliative strategies are recommended, ranging from chemotherapy, esophageal stents, brachytherapy (local radiotherapy), the surgical placement of jejunostomy or gastrostomy tubes, to esophageal bypass surgery [1,2,3,5]. In summary, treatment is typically multimodal, with neoadjuvant chemoradiotherapy followed by surgery showing the best overall cure rate and providing the best chance of a meaningful recovery [3,6,7,8]. Yet the optimal surgical approach (thoracic vs. transhiatal) and technique (open vs. minimally invasive) still have yet to be determined, with a need to clarify outcomes in terms of both survival and health-related quality of life [5]. 

Unfortunately, esophagectomies are associated with a high (5–10%) 90-day surgical mortality rate even at high-volume centers [5,9,10]. Furthermore, esophagectomies require a lengthy recovery process, which can be complicated by malnutrition, fatigue, anastomotic leaks, pneumonia, and recurrent laryngeal nerve damage [9,11,12]. In terms of quality of life outcomes, esophagectomies are related to poor voice outcomes, and rehabilitation may be challenging, at times requiring prostheses and procedural injections [13,14]. In fact, the risk of serious post-operative complications for all esophagectomy surgical approaches and techniques is 30% to 50% [5]. Likely because of the high rates of mortality and morbidity, studies have shown that esophagectomy is significantly underutilized, yet avoiding an appropriate esophagectomy is also associated with worse overall survival [8,15,16]. 

Clearly, there is still significant research needed to explore the high rates of complications and mortality, as well as possible interventions to mitigate these issues. On the other hand, there has been significant research into factors that impact esophageal cancer care. One disparity that has been well documented is that patients from rural areas tend to have worse survival rates [17,18], which is likely due to a multitude of reasons including socioeconomic status and the procedural volume of the treatment center. Procedural volume is significantly easier to characterize, with recent studies finding that patients who undergo esophagectomy at high-volume centers have better outcomes [19,20]. Even patients who travel significant distances to reach a high-volume center, compared with those who stay close to home at lower-volume centers, receive significantly different treatment options and have better outcomes overall [19,20,21]. Given this, there has been a push toward the regionalization of esophagectomies only to high-volume treatment centers in recent years with fewer opportunities for local care [22,23]. Typically, the individuals who need to travel significant distances are from more rural areas [24,25,26], which may also partially explain the rural versus urban disparity. While patients tend to do better at high-volume institutions, there are few studies evaluating the surgical outcomes and the recovery process of individuals who travel further or are from more rural locations. 

Gender disparities are also well characterized within esophageal cancer, both in terms of incidence and treatment. Esophageal cancer is a predominately male disease across the world, with a 4.2:1 male-to-female ratio in the United States alone [1,27,28]. In fact, the lifetime risk of esophageal cancer in the United States at about 1 in 127 in men and only 1 in 434 in women [4]. The etiology of incidence is likely multifactorial, including biological factors and sociocultural factors [27,29,30]. Biological factors include differences in the molecular subtype of cancer, as well as the role of estrogen exposure in women as a potential protective factor [27,30,31,32,33]. Culturally, men engage in higher-risk behaviors, such as consuming alcohol and smoking tobacco [29,34,35], which have been directly correlated to esophageal cancer rates. There has also been significant research into the role of gender in esophageal care, with recent studies showing that gender can act as an independent prognostic factor for overall prognosis [27,29,31,36,37,38]. While men might be at higher risk of esophageal cancer, studies have found that women receive lower rates of neoadjuvant treatment and surgery [37,39]. It is unclear as to the reasons why women are offered fewer treatment options, but these results are consistent with studies on several other illnesses, including heart disease, mental illness, and other cancers [37,40,41,42,43]. While a higher incidence of esophageal cancer in men has been clearly defined in the literature, there is little consensus on treatment outcomes between men and women. In fact, there have been multiple studies showing that women at times have better, worse, and the same overall survival compared with men [27,29,31,33,37]. Despite the research showing gender disparities in treatment, there has been less research into how gender plays a role in recovery after esophagectomy and after complications. 

Clearly, both gender and geographic access to care continue to play a large role in health disparities in esophageal cancer care despite advances in both surgical techniques and healthcare. The objective of our study is to evaluate differences in peri-operative outcomes following esophagectomy based on gender and geographic residence location in patients treated at a single high-volume esophageal cancer treatment center. We hypothesize that treatment at a high-volume center will eliminate gender and geographic disparities after esophagectomy. If there is variation, we hope to identify potential gaps in our cancer care paradigms. 

## 2. Materials and Methods

### 2.1. Patient Selection and Data Collection

A retrospective analysis of data collected for all patients receiving esophagectomy for loco-regional esophageal cancer (adenocarcinoma and squamous cell cancers) between January 2003 and December 2022 at a single tertiary care center was performed. Patients were identified by searching an electronic medical record database using relevant diagnosis and procedure codes. These individuals received their cancer care (including chemotherapy and radiation) from numerous different medical providers across the region but to be included in the study had to receive their surgery at a single, high-volume center. Patient demographics (age, gender, and race), comorbidities (cardiovascular diseases, diabetes, renal disease, and prior cancer), and medical history (BMI, smoking pack years, and substance use) were collected via chart review. The billing zip code was recorded for each patient for later use to determine the distance to a tertiary care center and state county. Any patient who traveled from out of state for surgery was excluded from the study. During the time period of the study, our medical center only recorded patients as male/female. Other data recorded, including details of surgery (both technique and approach), post-operative complications, length of stay, and 30-day mortality, were obtained from chart review.

### 2.2. Surgical Technique

The patients’ esophageal tumors were found in the middle third and the lower third and at the esophagogastric junction. Esophagectomies were performed via both open and minimally invasive surgical techniques depending on the individual’s case, tumor size, and anatomy. Surgical approaches favored the transhiatal approach but included the three-field and Ivor Lewis transthoracic approach. Minimally invasive techniques were both laparoscopic/thoracoscopic and robotic. 

### 2.3. Post-Operative Outcomes

Peri-operative data, including estimated blood loss, surgery duration, and blood product transfusions, were recorded for each patient. Data regarding post-operative recovery, complications, and hospital/intensive care unit (ICU) length of time were recorded. Key post-operative complications included pneumonia, acute respiratory distress syndrome (ARDS), pulmonary embolism, atrial arrhythmia requiring treatment, myocardial infarction, anastomotic leak, recurrent laryngeal nerve paresis, and chylothorax. The need for post-operative invasive procedures was also recorded, along with initial recovery in the ICU and the need for an additional visit to the ICU after their hospital course. Finally, mortality was recorded at discharge, 30 days, and greater than 30 days. 

### 2.4. Cohort Data Analysis

The whole cohort was divided based on gender (male and female), and the two groups were analyzed based on demographics, pre-operative treatment, surgical complications, post-operative outcomes, hospital length of stay (LOS), ICU LOS, and mortality. Chi-square analysis was used for categorical variables, including rates of comorbidities, use of neoadjuvant therapy (chemotherapy and radiation), post-operative complication rates (pneumonia, ARDS, pulmonary embolism, atrial arrhythmia requiring treatment, myocardial infraction, anastomotic leak, recurrent laryngeal nerve paresis, and chylothorax), and 30-day mortality rates. *t*-test analysis was completed for continuous variables including age, BMI, pack-year history, duration of surgery, and LOS.

The billing zip code was used to determine if the patient lived in a rural or metropolitan area. Each zip code was tied to a specific county in the state. Then, using the Rural Health Information Hub guidelines, the state counties were aggregated into existing state-level “metropolitan” or “rural” designations [44]. Metropolitan includes urban and suburban designations. The patient’s billing zip code was also used to help estimate the distance (in miles) they were required to drive to the hospital. First, the center radius of the zip code region was identified on a map; then, the driving distance to the hospital’s main entrance was calculated via Google Maps (accessed 3 May 2023 online). The distance traveled between the two groups was compared using *t*-test analysis. At this time, the whole cohort was divided based on home geographic location (rural and metropolitan). The metropolitan and rural groups were then analyzed based on the same parameters as the gender groups (see above) using chi-squared and paired *t*-tests. 

Finally, the cohort was analyzed based on both gender and geographical location. The “male” and “female” groups were divided based on geographical location into four different groups: (1) rural females, (2) metropolitan females, (3) rural males, and (4) metropolitan males. The four groups were further analyzed based on the above criteria using single-factor ANOVA to compare the groups and *t*-test analysis. 

## 3. Results

### 3.1. Patient Characteristics 

A total of 1716 patients underwent an esophagectomy between January 2002 and December 2022, with 1545 patients meeting the criteria to be in the study. The majority of operations were performed “open” (84%) when compared with minimally invasive operations (16%). Furthermore, the majority of the “open” operations were transhiatal esophagectomies (55%). Of note, there was no difference in approach to surgery when analyzed based on gender or geographic home location. 

#### 3.1.1. Analysis of Demographics, Comorbidities, and Neoadjuvant Treatment Based on Gender

Of all the patients, 83.6% (1292/1545) identified as male, and 16.4% (253/1545) identified as female. There was no significant difference in age between men and women, with men being 63.8 ± 9.9 years of age and women being 64 ± 11.2 years of age (*p* = 0.84). The men had a significantly higher BMI (*p* = 0.009) at 29.39 ± 10.23 kg/m^2^, while the women’s average BMI was 27.63 ± 7.55 kg/m^2^. Men also had a greater number of pack-years at 29.4 ± 10.2 pack-years compared with 21.7 ± 25.7 pack-years for women (*p* < 0.001). 

In terms of comorbidities, men were more likely to suffer from coronary artery disease (CAD) at 19.3% compared with women at 11.5% (*p* = 0.002). There were no significant differences in other comorbidities, including congestive heart failure (CHF), diabetes, and hypertension (Table 1). 

Regarding neoadjuvant therapy, men were significantly more likely to have undergone both pre-operative chemotherapy (*p* = 0.002) and pre-operative radiation (*p* = 0.003), with 60.4% of men, as compared with 49.8% of women, undergoing pre-operative chemotherapy and 59.3% (781/1292) of men as compared with 49.4% of women undergoing pre-operative radiation. Please see Table 1.

#### 3.1.2. Analysis of Demographics, Comorbidities, and Neoadjuvant Treatment Based on Geographic Residence Location

The patients were sorted into appropriate state counties using their reported billing zip codes. Based on Rural Health Information guidelines, each patient was then sorted into previously existing state-level “metropolitan” and “rural” designations, and then, the driving distance to the hospital was calculated. Rural patients traveled greater distances for healthcare services compared with metropolitan patients (rural: 192.4 ± 101.2 miles; metropolitan: 63.4 ± 36.7 miles; *p* < 0.001). 

Analysis of demographics showed that, of the total cohort, 76.0% (1174/1545) of patients were from a metropolitan area, and 24.0% (371/1545) were from a rural area. There was no significant difference in age between the metropolitan and rural groups. There were also no differences in average BMI or number of pack-years between the two groups. 

In terms of comorbidities, there were no differences between the groups or in rates of neoadjuvant treatment with chemotherapy and radiation. Please see Table 2. 

#### 3.1.3. Analysis of Demographics, Comorbidities, and Neoadjuvant Treatment Based on Gender and Geographic Residence Location 

When separated by both geographic criteria and gender, two differences are readily apparent: (1) rural men had the highest number of pack-years, especially compared with rural females (*p* = 0.002), and (2) metropolitan women had the lowest rates of CAD, especially compared with rural women (*p* < 0.001). 

In terms of neoadjuvant therapy, metropolitan women were outliers, with significantly lower rates of pre-operative chemotherapy (*p* = 0.006) and lower rates of pre-operative radiation (*p* = 0.025). Please see Table 3 and Table 4.

### 3.2. Peri-Operative Outcomes

#### 3.2.1. Intra-Operative Findings, Post-Operative Complications, and Inpatient Recovery Times Based on Gender

Women had significantly longer surgeries compared with men (*p* < 0.001), with no other intra-operative differences. 

Post-operatively, men experienced higher rates of recurrent laryngeal nerve paresis at 5.3% compared with 1.2% of women (*p* = 0.004). On the other hand, women were more likely to experience chylothorax complications at 8.7% compared with 4.7% of men (*p* = 0.01). There was no difference between men and women in other post-operative complications. Please see Table 5. 

In terms of recovery time, women had significantly longer hospital stays compared with men (*p* = 0.002). Women also had significantly longer stays in the ICU compared with men (*p* = 0.03). There were no differences in ICU visits and no differences in 30-day mortality between the two genders. Please see Table 5. 

#### 3.2.2. Intra-Operative Findings, Post-Operative Complications, and Inpatient Recovery Times Based on Geographic Residence Location 

Intra-operatively, there was no difference between the rural and metropolitan groups for surgical duration and need for blood transfusion. 

In terms of post-operative complications, rural patients had higher rates of pneumonia at 8.1% compared with 5.3% metropolitan patients (*p* = 0.04), with no other significant differences in other post-operative complications. 

In terms of post-operative recovery time, there were no differences between rural and metropolitan patients in hospital LOS, but rural patients had more visits to the ICU (rural: 11.6% (43/371); metropolitan: 8.1% (94/1174); *p* = 0.03) and longer LOSs in the ICU (rural: 1.59 ± 6.96 days; metropolitan: 0.65 ± 3.13 days; *p* < 0.001). Of note, there was no difference in 30-day mortality between the rural and metropolitan groups. Please see Table 6.

#### 3.2.3. Intra-Operative Findings, Post-Operative Complications, and Inpatient Recovery Times Based on Both Gender and Geographic Location

When separated by both geographic criteria and gender, there were no significant differences in intra-operative findings between the four groups. Interestingly, both rural and metropolitan males were more likely to experience recurrent laryngeal nerve paresis compared with their female counterparts. Of note, there were no significant differences in rates of recurrent laryngeal nerve paresis between rural and metropolitan males, signifying that, within this cohort, men were significantly more likely than women to suffer from recurrent laryngeal nerve paresis regardless of geographic residence. 

Analysis of inpatient recovery time unveiled that rural women were outliers, with significantly longer hospital LOSs (*p* < 0.001), higher rates of ICU admission (*p* < 0.001), and longer ICU LOSs (*p* < 0.001) as compared with their metropolitan and male counterparts. Interestingly, rural men had similar overall LOSs and ICU visits compared with metropolitan men and women. There were no differences in 30-day mortality between all four groups. Please see Table 4 and Table 7. 

## 4. Discussion

In summary, we found that there were significant health disparities based on gender and geographic home location despite all surgeries being performed at the same high-volume esophageal cancer center. More specifically, the disparities we found were as follows: (1) women and rural patients, especially rural women, had a more complex inpatient recovery process with longer LOSs both in the hospital and ICU; (2) post-operatively, men had higher rates of recurrent laryngeal nerve paresis, while women had higher rates of post-operative chylothoraxes; and (3) rural patients had higher rates of pneumonia rates compared with metropolitan counterparts. Given that there were minimal differences in demographics and pre-surgical medical history, there is no clear reason for these findings based solely on expected outcomes. Furthermore, while pre-operative care was at a multitude of different medical centers, all surgeries were performed at a single center under the same guidelines and post-operative care. Without any significant differences prior to surgery, it is surprising that rural patients and women had worse surgical outcomes. 

Our cohort is not the first study to show that women or rural patients can have worse outcomes after a major surgery, with previous studies showing that women experience numerous barriers to care and recovery. First, as found in the literature, as well as confirmed by our cohort analysis, women receive significantly lower rates of neoadjuvant treatment [37,39]. Yet while women are less likely to receive certain treatments, there is less clarity in the literature on the overall survival and recovery after esophagectomy, with multiple studies showing that women at times have better, worse, and the same overall survival compared with men [27,29,31,33,37]. While our cohort showed no differences in overall survival, we did show that women have a more complex recovery process after esophagectomy, specifically a longer hospital LOS and a longer ICU LOS. While gender disparities in recovery after esophagectomy are not well defined in the literature, prolonged recovery processes for women have been well characterized in numerous cardiac surgery studies [45,46,47], but we are one of a few studies to highlight worse recovery after esophagectomy. Of note, the male and female cohorts were similar, with men having a higher BMI, a greater number of pack-years, and higher rates of CAD, all of which are consistent with national gender trends [34,48,49] and do not lend any clarity to why women have a prolonged recovery. Interestingly, men were more likely to have pre-operative rates of chemotherapy and radiation, which often results in patients being weaker when they receive surgery. This conflicts with the prolonged recovery course seen in women.

Similarly to gender, geographic access to care has played a large role in health disparities, with previous studies showing that rural individuals tend to have worse survival rates [17,18] for unclear reasons. While our rural cohort did not have worse overall survival, we did find that rural patients had significantly longer inpatient recovery times as compared with their metropolitan counterparts, despite both groups being similar in terms of demographics and comorbidities. Again, there is no clear reason for these findings, and further research needs to be conducted to identify why rural patients have a more complex recovery process. One possible theory for these prolonged recovery processes is that rural patients typically need to travel greater distances to access healthcare services, including pre-operative surgical assessment, surgery, and even their primary oncologists [22,23,24]. Unsurprisingly, in our cohort, rural individuals needed to travel significantly greater distances to reach the tertiary surgical care center, signaling that this distance barrier may have led to less extensive pre-operative workups and less access to local support. This may have led to the prolonged inpatient recovery, but our thoughts are purely speculative and based on anecdotal patient care.

When we analyzed the groups based on both gender and geographic residence location, we saw that rural women specifically tended to be outliers, with more ICU visits and longer hospital LOSs. Interestingly, rural women were not stark outliers in the demographics—in fact, rural women had the fewest number of pack-years of tobacco smoking. Of note, rural women were the most likely to have CAD but did not have higher rates of post-operative complications. We wondered if rates of neoadjuvant therapy affected these complications and ICU stays, but it was metropolitan women who were the outliers in terms of pre-operative chemoradiotherapy compared with their rural and male counterparts. In fact, rural women received neoadjuvant treatments at similar rates to rural men and metropolitan men. At this time, it is still unclear why rural women had a prolonged inpatient recovery process, which highlights an area of medicine that requires more research. One possible reason for the longer recovery process is that rural women may be the most vulnerable group, suffering from both a distance barrier and gender disparities, with less extensive workups, less access to care, and less supportive care at home.

Briefly, in terms of post-operative complications, we saw that men were more likely to have recurrent laryngeal nerve paresis, while women’s surgeries were more likely complicated by chylothoraxes. One possible explanation for increased rates of chylothoraxes in women is patient size, which could impact the rate of thoracic duct disruption caused by the blind mediastinal dissection, resulting in higher rates of chylothoraxes. Overall, these differences are unclear and puzzling, as demographics and pre-operative differences were minimal between the groups, with none of the literature describing gender differences in these specific complication rates. Furthermore, rural patients in our cohort were more likely to suffer from pneumonia, while there were no differences when broken down by both gender and location. While higher rates of pneumonia in rural patients have not been described in the literature, one possible cause in our cohort is that the rural patients had a significant history of smoking, with more pack-years reported than the metropolitan group. This greater history of smoking can increase the risk of pulmonary complications, including pneumonia [50]. Finally, rural patients, as mentioned above, may suffer from a distance barrier, leading to less extensive workups and worse access to care.

While this study shows that health disparities are pervasive in both gender and geographic home location, we recognize that there are other factors that may influence the trends and are potential limitations to our study. First, the sample size of the female cohort was very small compared with the male cohort (although it was similar to the national gender ratio of men to women with esophageal cancer). Similarly, the rural cohort was significantly smaller than the metropolitan cohort. The small nature of these cohorts, especially the fact that the rural women cohort was the smallest at 43 of the 1545 total patients, may influence some of the trends we see, and further studies will need to be conducted to ensure that this trend is still seen within a larger cohort. Secondly, the retrospective design of our study limits some of our available data. For example, while all surgeries were completed at the same institution, the pre-operative evaluations and treatments were completed at multiple different clinics and with multiple providers throughout the state. This means that there could be differences in numerous confounding factors that we were unable to control for, including, but not limited to, neoadjuvant therapy in dosage, the type of chemotherapy, the period between neoadjuvant therapy and surgery, radiation dosages, and pre-operative workups. This may also partially explain the different trends we see for neoadjuvant therapies with women receiving lower rates of pre-operative chemotherapy and radiation. This, in turn, may have influenced a patient’s frailty prior to surgery and the specific timing of recovery. Finally, two important parameters that influence all complications, serum hemoglobin and albumin levels, were not collected in this study. It is possible that there was a significant difference in these serum levels between men and women, which may partially explain some of the differences we saw in post-operative complications [51,52,53,54,55].

Regardless of the limitations of this study, we cannot ignore the significant trend showing that rural women had a significantly longer inpatient recovery process after esophagectomy compared with female metropolitan or male counterparts despite receiving surgery at the same institution. Given these significant disparities in care for rural women, it is pertinent to implement more resources for rural women, perhaps for overall health, but specifically in the setting of esophageal cancer and preparing for surgery. We recommend targeting pre-operative optimization for rural female patients to help combat these disparities and improve inpatient recovery. While better pre-operative assessments are clearly needed for rural women, we also recommend enhanced post-operative care assessments for both women and rural patients as well. Furthermore, in recent years, there has been a greater push for a more personalized approach to cancer treatment [27,56], and within this framework. we recommend that both gender and geographic home location be considered as unique factors in the treatment course and prognosis of esophageal cancer. We also believe there needs to be more research on the root causes of disparities in gender and geographic location, possibly with qualitative research to recognize what patients perceive as barriers to care and how healthcare providers can improve both pre-operative and post-operative care. Finally, once interventions are implemented, research is needed to focus on the impact of pre- and post-operative service assessments in eliminating disparities for rural-based females undergoing esophagectomy.

## 5. Conclusions

In this retrospective review, we found that both women and rural patients had a prolonged recovery process after esophagectomy despite receiving surgery at the same high-volume institution. When further broken down by both geographic location and gender, rural women were especially vulnerable, with longer hospital and ICU LOSs. It remains unclear why rural women experience prolonged post-operative recovery after esophagectomy, but this highlights a need for both tailored pre-operative and post-operative interventions in this population. We recommend that healthcare providers take a more personalized approach to care and account for gender and geography in planning treatment options. Future research is warranted to confirm these findings and further evaluate contributing factors, thus improving the access to care and surgical outcomes of rural women. 

## Figures and Tables

**Table 1 cancers-16-01078-t001:** Demographics, comorbidities, and neoadjuvant treatment based on gender. (* indicates *p* < 0.05).

	Female (n = 253)	Male (n = 1292)	*p*-Value
Age	64 ± 11.2	63.8 ± 9.9	0.84
BMI *	27.63 ± 7.55	29.39 ± 10.23	0.009
Pack-years *	21.7 ± 25.7	28.9 ± 29.1	<0.001
Comorbidities			
CAD *	11.5% (29)	19.3% (250)	0.002
CHF	2.4% (6)	2.7% (35)	0.76
Diabetes	19.8% (50)	19.4% (246)	0.79
HTN	55.3% (140)	51.3% (663)	0.24
Pre-op chemo *	49.8% (126)	60.4% (781)	0.002
Pre-op radiation *	49.4% (125)	59.3% (766)	0.003

**Table 2 cancers-16-01078-t002:** Demographics, comorbidities, and neoadjuvant treatment based on geographic home location. (* indicates *p* < 0.05).

	Metropolitan (n = 1174)	Rural (n = 371)	*p*-Value
Age	63.8 ± 10.2	64 ± 10.5	0.43
BMI	28.97 ± 7.44	29.5 ± 15.14	0.38
Pack-years *	21.7 ± 25.7	28.9 ± 29.1	<0.001
Comorbidities			
CAD	17.0% (199)	21.6% (80)	0.05
CHF	2.47% (29)	3.23% (12)	0.43
Diabetes	18.5% (217)	21.3% (79)	0.24
HTN	52.5% (616)	50.4% (187)	0.24
Pre-op chemo	57.6% (676)	62.3% (231)	0.11
Pre-op radiation	57.2% (672)	59.0% (219)	0.58

**Table 3 cancers-16-01078-t003:** Demographics, comorbidities, and neoadjuvant treatment based on both gender and geographic home location. (* indicates *p* < 0.05).

	Female Metro (n = 210)	Female Rural (n = 43)	Male Metro (n = 964)	Male Rural (n = 328)	*p*-Value
Age	64.3 ± 11.2	62.8 ± 11	63.7 ± 9.9	64.4 ± 9.9	0.54
BMI	27.77 ± 7.73	26.95 ± 6.67	29.24 ± 7.35	29.82 ± 15.90	0.06
Pack-years *	27.8 ± 25.6	22.5 ± 26.3	28.4 ± 28.9	30.3 ± 29.5	0.002
Comorbidities					
CAD *	8.1% (17)	27.9% (12)	18.9% (182)	20.7% (68)	<0.001
CHF	2.4% (5)	2.3% (1)	2.5% (24)	3.4% (11)	0.85
Diabetes	18.1% (38)	27.9% (12)	18.6% (179)	20.4% (67)	0.42
HTN	55.7% (117)	53.4% (23)	51.2% (499)	50% (164)	0.62
Pre-op chemo *	48.1% (101)	58.1% (25)	59.6% (575)	62.8% (206)	0.006
Pre-op radiation *	48.1% (101)	55.8% (24)	59.2% (571)	59.5% (195)	0.025

**Table 4 cancers-16-01078-t004:** *t*-test analysis comparing the four groups based on both gender and geographic location (* indicates *p* < 0.05).

	Female Metro: Female Rural	Female Metro: Male Metro	Female Metro: Male Rural	Female Rural: Male Metro	Female Rural: Male Rural	Male Metro: Male Rural
CAD	<0.001 *	<0.001 *	<0.001 *	0.07	<0.001 *	0.15
Pre-op chemo	0.002 *	0.001 *	<0.001 *	0.55	0.13	0.11
Pre-op radiation	0.005 *	0.003 *	0.004 *	0.31	0.49	0.24
RLN paralysis	0.008 *	0.006 *	0.01 *	0.21	0.38	0.25
Initial visit to ICU	<0.001 *	<0.001 *	<0.001 *	<0.001 *	<0.001 *	<0.001 *
Hospital LOS	0.03 *	0.09	0.23	<0.001 *	<0.001 *	0.73
ICU LOS	0.23	0.08	0.52	0.12	0.31	0.02 *

**Table 5 cancers-16-01078-t005:** Peri-operative outcomes based on gender. (* indicates *p* < 0.05).

	Female (n = 253)	Male (n = 1292)	*p*-Value
Blood transfusion	6.2% (13)	3.2% (41)	0.12
Surgery duration (min) *	340 ± 103.27	263.36 ± 227.15	<0.001
Post-op events	75.5% (191)	71.5% (924)	0.19
Pneumonia	3.9% (10)	6.3% (82)	0.14
ARDS	2.0% (5)	1.7% (22)	0.76
PE	1.6% (4)	1.2% (16)	0.66
Atrial arrhythmia requiring treatment	17.4% (44)	20.7% (268)	0.22
MI	0.0% (0)	0.3% (4)	0.44
Anastomotic leak	19.8% (50)	16.6% (214)	0.22
Recurrent laryngeal nerve paresis *	1.2% (3)	5.3% (68)	0.004
Chylothorax	8.7% (22)	4.7% (61)	0.01
Post-op invasive procedure	16.6% (42)	14.2% (183)	0.31
Hospital LOS *	13.19 ± 12.56	11.18 ± 9.42	0.002
Initial visit to ICU	11.1% (28)	8.4% (109)	0.18
Additional visit to ICU	5.1% (13)	3.8% (49)	0.32
ICU LOS *	1.43 ± 5.74	0.77 ± 4.06	0.03
Mortality			
At discharge	1.6% (4)	0.9% (12)	0.34
30 days	1.6% (4)	1.0% (13)	0.42
Long term	14.2% (36)	14.2% (183)	0.98

**Table 6 cancers-16-01078-t006:** Pre-operative outcomes based on geographic residence location. (* indicates *p* < 0.05).

	Metropolitan (n = 1174)	Rural (n = 371)	*p*-Value
Blood transfusion	3.9% (46)	2.4% (9)	0.17
Surgery duration (min)	372.84 ± 161.25	363.1 ± 102.78	0.33
Post-op events	71.0% (834)	75.7% (281)	0.09
Pneumonia *	5.3% (62)	8.1% (30)	0.04
ARDS	1.5% (18)	2.4% (9)	0.25
PE	1.4% (16)	1.1% (4)	0.67
Atrial arrhythmia requiring treatment	19.4% (227)	22.9% (85)	0.14
MI	0.26% (3)	0.27% (1)	0.97
Anastomotic leak	17.3% (203)	16.4% (61)	0.70
Recurrent laryngeal nerve paresis	4.5% (53)	4.8% (18)	0.8
Chylothorax	5.5% (64)	5.1% (19)	0.79
Post-op invasive procedure	13.8% (162)	17.0% (63)	0.14
Hospital LOS	11.36 ± 10.00	12.00 ± 10.10	0.28
Initial Visit to ICU *	8.1% (94)	11.6% (43)	0.03
Additional visit to ICU	4.1% (48)	3.7% (14)	0.79
ICU LOS *	0.65 ± 3.13	1.59 ± 6.96	<0.001
Mortality			
At discharge	1.02% (12)	1.1% (4)	0.92
30 days	1.02% (12)	1.6% (6)	0.35
Long-term	14.4% (169)	13.4% (50)	0.64

**Table 7 cancers-16-01078-t007:** Pre-operative outcomes based on gender and geographic home location (* indicates *p* < 0.05).

	Female Metro (n = 210)	Female Rural (n = 43)	Male Metro (n = 964)	Male Rural (n = 328)	*p*-Value
Blood transfusion	6.2% (13)	2.4% (1)	3.4% (33)	2.4% (8)	0.13
Duration	345.02 ± 106.37	322.17 ± 83.07	379.18 ± 170.40	369.58 ± 103.87	0.11
Post-op events	74.7% (157)	79.1% (34)	70.2% (677)	83.5% (247)	0.12
Pneumonia	3.3% (7)	7.0% (3)	5.7% (55)	8.2% (27)	0.12
ARDS	2.4% (5)	0.0% (0)	1.3% (13)	2.7% (9)	0.26
PE	1.4% (3)	2.4% (1)	1.3% (13)	0.9% (3)	0.85
Atrial arrhythmia requiring treatment	14.8% (31)	30.2% (13)	20.3% (196)	21.9% (72)	0.07
MI	0.0% (0)	0.0% (0)	0.3% (3)	0.3% (1)	0.85
Anastomotic leak	19.5% (41)	20.0% (9)	16.8% (162)	15.8% (52)	0.63
Recurrent laryngeal nerve paresis *	1.0% (2)	2.4% (1)	5.3% (51)	5.2% (17)	0.04
Chylothorax	8.1% (17)	11.6% (5)	4.9% (47)	4.3% (14)	0.054
Post-op invasive procedure	15.2% (32)	23.3% (10)	13.5% (130)	16.2% (53)	0.23
Hospital LOS *	12.40 ± 11.64	17.02 ± 15.85	11.13 ± 9.60	11.34 ± 8.88	0.001
Initial visit to ICU *	9.5% (20)	18.6% (8)	7.7% (74)	10.6% (35)	<0.001
Additional visit to ICU	4.8% (10)	7.0% (3)	3.9% (38)	3.4% (11)	0.64
ICU LOS *	1.10 ± 4.30	3.02 ± 10.10	0.55 ± 2.80	1.40 ± 6.43	<0.001
Mortality					
At discharge	1.4% (3)	2.4% (1)	0.9% (9)	0.9% (3)	0.76
30 days	1.4% (3)	4.7% (2)	0.9% (9)	1.2% (4)	0.16
Long-term	13.3% (28)	18.6% (8)	14.6% (141)	12.8% (42)	0.68

## Data Availability

Restrictions apply to the datasets. The data can be available after a data agreement form is completed between institutions.

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
