# Peer review of "Rural Women Have a Prolonged Recovery Process after Esophagectomy"

_cancers, 2024, doi:10.3390/cancers16061078_

Round 1

Reviewer 1 Report

Comments and Suggestions for Authors

Abstract:

- The abstract should include a clear objective statement identifying the purpose of the study and specific aims. For example: "The objective of this study was to evaluate disparities in outcomes of esophageal cancer surgery based on gender and geographic location."

- In the methods, briefly state that this was a retrospective cohort study and provide the number of patients included.

- The results should highlight only the most important 1-2 findings rather than listing many statistics. Focus on the key disparities found.

- Add a conclusion statement summarizing the main implications of the study, such as: "Rural female patients undergoing esophagectomy experienced worse postoperative outcomes compared to other groups, suggesting a need for targeted interventions in this population."

Introduction:

- Provide more background on esophageal cancer incidence, risks, and current treatment paradigms. Describe role of surgery and typical outcomes.

- line 52, esophagectomy are related to poor voice outcomes and rehabilitation may be challenging. discuss and cite PMID: 28958139.

- Introduce the topic of healthcare disparities and what is known from prior literature about differences in esophageal cancer care by gender and geographic location. Highlight the role on environmental treatment effect.

- Clearly state the rationale and objectives for the study: "Despite advances in surgical techniques, disparities may persist in outcomes for certain subgroups of patients. This study aimed to evaluate differences in perioperative outcomes following esophagectomy based on gender and geographic residence location in patients treated at a high-volume center."

- End the introduction with a statement of the study objectives/aims.

Methods:

- Provide more details on the data source and cohort selection criteria. For example: "This was a retrospective analysis of patients undergoing esophagectomy for esophageal cancer at University Hospital from 2010-2020. Patients were identified by searching the electronic medical record database using relevant diagnosis and procedure codes."

- Describe what specific data was extracted from records: "Patient demographics, comorbidities, details of surgery, postoperative complications, length of stay, and 30-day mortality were obtained from chart review."

- Elaborate on the statistical analysis: "Characteristics and outcomes were compared between groups using chi-square tests for categorical variables and t-tests for continuous variables."

Results:

- Focus the results on the most important findings that directly relate to the study objectives stated in the introduction. Move secondary analyses to supplementary data.

- Present key results in a visual format such as graphs or summary tables where appropriate.

- Structure the results by specific objective rather than just gender and location. For example:

1. Postoperative complications by gender
2. ICU length of stay by location
3. Outcomes based on gender and location

- Use headings and transition sentences to guide the reader through the findings.

- Do not just present data, interpret the results - what do they mean in the context of the study goals? Why are certain results unexpected or significant?

Discussion:

- Interpret and contextualize the key findings, including:

1) Rural women had worse postoperative outcomes compared to other groups
2) Men had higher rates of recurrent laryngeal nerve paresis
3) Rural patients had higher pneumonia rates  

- For each key finding, discuss potential reasons based on prior literature and mechanisms. Provide context for why these disparities matter.

- Acknowledge limitations more thoroughly:

- Small sample sizes for rural and female subgroups
- Retrospective design limits available data
- Unable to account for potential confounding factors

- Compare results to prior studies on this topic and explain similarities/differences.

- Discuss the implications of the results - do they suggest a need for:

1) Targeted preoperative optimization of rural female patients?
2) Enhanced postoperative support for rural patients?
3) Further research on causes of disparities?

Conclusions:

- Restate the main study findings concisely

- Emphasize the real-world significance of the results

- For example: "Rural female patients experienced worse postoperative recovery after esophagectomy compared to other groups, highlighting a need for tailored preoperative and postoperative interventions in this population. Additional research is warranted to confirm these findings and further evaluate contributing factors."

- Avoid overstating conclusions beyond what can be supported by the data

Comments on the Quality of English Language

minor errors

Author Response

Comments (with responses below):

Reviewer #1:

Abstract:
1. The abstract should include a clear objective statement identifying the purpose of the study and specific aims. For example: "The objective of this study was to evaluate disparities in outcomes of esophageal cancer surgery based on gender and geographic location."

Response: We appreciate the suggestion to add an objective statement to the abstract, and agree it is helpful to the reader. The abstract has been edited to include the statement “The aim of our study was to evaluate for disparities in outcomes for patients undergoing esophagectomy based on gender and geographic location.” and has been edited slightly to keep it under the 200 words abstract limit.

2. In the methods, briefly state that this was a retrospective cohort study and provide the number of patients included.

Response: We appreciate the comment. We routinely do no put the number of patients in the methods and have kept that in the Results section.  The data was collected prospectively also.  The abstract has been altered accordingly to incorporate this helpful comment using the phrase that “A retrospective cohort of prospectively collected perioperative data from patients who underwent esophagectomy from 2003 to 2022 was identified”.

3. The results should highlight only the most important 1-2 findings rather than listing many statistics. Focus on the key disparities found.

Response: Thank you for the suggestion. We have modeled our abstract as per preferences for surgical journals. The results have been edited to focus on two key disparities: 1) that women have more complex recovery processes after esophagectomy compared to their male counterparts; and 2) when further divided based on geographic location, rural women are further outliers compared to their metropolitan female and male counterparts.

  1. Add a conclusion statement summarizing the main implications of the study, such as: "Rural female patients undergoing esophagectomy experienced worse postoperative outcomes compared to other groups, suggesting a need for targeted interventions in this population."

Response: We appreciate this suggestion and agree that this is helpful to provide a more succent abstract that highlights the major finding in our manuscript. It also helps highlight the key results that we decided to focus on in comment #3. The conclusion has been rewritten as follows: “Rural female patients undergoing esophagectomy were more likely to have a longer inpatient recovery process compared to their female metropolitan or male counterparts, suggesting a need for more targeted interventions in this population.”

Introduction:
5. Provide more background on esophageal cancer incidence, risks, and current treatment paradigms. Describe role of surgery and typical outcomes.

Response: We appreciate the comments and have added some more background on esophageal cancer (please see highlighted sections of the introduction). In terms of background on incidence, we have added information on projected incidence and death rates for 2024, as well as the different histological subtypes of esophageal cancer and the world distribution. Risk factors were correlated to the increase rates of adenocarcinoma we are seeing in the Western world, but other major risk factors of esophageal cancer were also discussed. Finally, further information regarding current treatment paradigms were expanded upon, and the typical surgical outcomes were discussed further.

  1. line 52, esophagectomy are related to poor voice outcomes and rehabilitation may be challenging. discuss and cite PMID: 28958139.

Response: Thank you for emphasizing the importance of quality-of-life complications as well after esophagectomies. The poor voice outcomes and the complicated rehabilitation has been discussed, with the article suggested cited, as well as a second. 

  1. Introduce the topic of healthcare disparities and what is known from prior literature about differences in esophageal cancer care by gender and geographic location. Highlight the role on environmental treatment effect.

Response: We appreciate the comment. The manuscript has been altered to include a more clear and concise description of healthcare disparities in esophageal care regarding gender and geographic location. We have also added some better transitions to ensure that this is clear to the reader. We are unclear as to the reviewer’s comments on environmental treatment effect.

  1. Clearly state the rationale and objectives for the study: "Despite advances in surgical techniques, disparities may persist in outcomes for certain subgroups of patients. This study aimed to evaluate differences in perioperative outcomes following esophagectomy based on gender and geographic residence location in patients treated at a high-volume center."

Response: We appreciate the comments. While the objective was previously stated, the rationale for why our study was key was not as clear. We added a clear rationale before objective statement that should explain the importance of our study more. Our original rationale/objective for the study was “The goal of our study is to examine the gender and geographic disparities surrounding the inpatient recovery process after esophagectomy at a single high-volume esophageal cancer treatment center”, which was slightly altered for enhancement to “The objective of our study is to evaluate differences in perioperative outcomes following esophagectomy based on gender and geographic residence location in patients treated at a single high-volume esophageal cancer treatment center”.

9. End the introduction with a statement of the study objectives/aims.

Response: While we appreciate the desire to end the introduction with the study’s objectives/aims, we feel that our current conclusion is much more logical after editing. After discussion between the authors, we feel that it was helpful to provide the audience with our hypothesize after the goal of our study to partially explain our thought process.

Methods:

10. Provide more details on the data source and cohort selection criteria. For example: "This was a retrospective analysis of patients undergoing esophagectomy for esophageal cancer at University Hospital from 2010-2020. Patients were identified by searching the electronic medical record database using relevant diagnosis and procedure codes."

Response: Further details were added by the authors to provide more information on data sourcing and exclusion/inclusion criteria. Thank you  

11. Describe what specific data was extracted from records: "Patient demographics, comorbidities, details of surgery, postoperative complications, length of stay, and 30-day mortality were obtained from chart review."

Response: We appreciate the suggestion. We provided further details and information for each section. Of note, “details of surgery, postoperative complications, length of stay, and 30-day mortality” were previously provided, just in separate sections.

12. Elaborate on the statistical analysis: "Characteristics and outcomes were compared between groups using chi-square tests for categorical variables and t-tests for continuous variables."

Response: Thank you for your comments. We provided further elaboration on the statistical analysis that should help understand the thought process. 

Results:

13. Focus the results on the most important findings that directly relate to the study objectives stated in the introduction. Move secondary analyses to supplementary data.

Response: We appreciate the comments. We’d favor keeping the tables as is with the secondary analyses present unless there are space concerns.

14. Present key results in a visual format such as graphs or summary tables where appropriate.

Response: We appreciate the suggestion. We have removed some of the data from text and referred to it in our summary tables. Of note, we have also highlighted the key results in the table using “*” to signify importance.

  1. Structure the results by specific objective rather than just gender and location. For example:

    1. Postoperative complications by gender
    2. ICU length of stay by location
    3. Outcomes based on gender and location

Response: We appreciate the reviewer’s comments.  We would favor keeping the results section as currently structured.  We recognize that there are many ways to structure the results.  The way we have thought about the data and presented it in the tables, we’d keep the structure as is.

16. Use headings and transition sentences to guide the reader through the findings.

Response: We appreciate the comments and agree that better headings could help guide the reader through the findings in a more efficient manner. We have also added numerous transition sentences throughout the results section to help with the flow, please see the highlighted parts of the result section. 

17. Do not just present data, interpret the results - what do they mean in the context of the study goals? Why are certain results unexpected or significant?

Response: While we appreciate the suggestion, we feel that the results section should be limited to objective data presentation. We have purposefully limited the amount of analysis and interpretation in the results section to ensure less bias and provide our readers the opportunity to critically analyze the data without input from us, the authors. We do agree that these results need to be interpreted slightly clearer in the context of the study and why certain results were unexpected or significant. Therefore, we have re-worked the beginning of the discussion section to better interpret the results and place it in the appropriate context. Please see the highlighted portions of that section.

Discussion:

18. Interpret and contextualize the key findings, including:
1) Rural women had worse postoperative outcomes compared to other groups
2) Men had higher rates of recurrent laryngeal nerve paresis
3) Rural patients had higher pneumonia rates  

Response: We appreciate the comments and have added a nice summary off these key findings at the beginning of the discussion section (please see the highlighted section). We have also briefly interpreted and contextualized these key findings, and later generalized in the broader context of the literature.

  1. For each key finding, discuss potential reasons based on prior literature and mechanisms. Provide context for why these disparities matter.

Response: We have correlated our key findings to prior literature and have also spent more time talking about the potential reasons and possible theories based on both previous studies and our own interpretation of the results. Please see the highlighted portion in paragraphs 2, 3 and 4 for significant edits. We appreciate the input and belief it has helped enhance our discussion and conclusions.

20. Acknowledge limitations more thoroughly:
- Small sample sizes for rural and female subgroups
- Retrospective design limits available data
- Unable to account for potential confounding factors

Response: Thank you for your suggestion. After review, we agree that the limitations of the study were not as thoroughly acknowledged as they could have been. We ensured that the three limits you recommended above were acknowledged and added more details for others. Finally, reviewer #2 helped us to identify a potential bias, which has also been added to the limitation section.

  1. Compare results to prior studies on this topic and explain similarities/differences.

Response: Please see response to #19 and see highlighted portions of text to see our approach to responding to this comment.

22. Discuss the implications of the results - do they suggest a need for:
1) Targeted preoperative optimization of rural female patients?
2) Enhanced postoperative support for rural patients?
3) Further research on causes of disparities?

Response: We appreciate the comments and while we had briefly touched on the implications of the results, we agree that there were areas to expand and further emphasize the important implications of our results. In our final paragraph of the discussion section we added 3 different sections that should address this comment.

Conclusions:

23. Restate the main study findings concisely

Response: We concisely restated the main study findings, focusing on our major conclusion that rural women as substantially longer LOS in the hospital and ICUs. We also stated that there are unclear reasons for why these disparities exist.

24. Emphasize the real-world significance of the results
- For example: "Rural female patients experienced worse postoperative recovery after esophagectomy compared to other groups, highlighting a need for tailored preoperative and postoperative interventions in this population. Additional research is warranted to confirm these findings and further evaluate contributing factors."

Response: We appreciate the comment and have added 2 sentences within our conclusions that emphasize the real-world significance of our results. Please see highlighted sections of the conclusion that highlight the need for interventions for rural women and the need for additional research. 

25. Avoid overstating conclusions beyond what can be supported by the data

Response: We appreciate the comments and believe we have not overstated our conclusions, emphasizing the need for research to confirm these findings as well as highlight the importance of following up to discover the root causes of these issues.

Reviewer 2 Report

Comments and Suggestions for Authors

Thank you for the opportunity to review this important study. Here are my comments and suggestions.

Line 90: The study is either retrospective or prospective. Even in prospective studies, the results are analyzed ''retrospectively'' because they are analyzed at the end of the prospective period. The main difference between retrospective and prospective studies is that in retrospective studies, the patients went through institutional protocol, and when the idea for the study came out, there was a possibility that there was not all the necessary data in the electronic medical records. In a prospective study, the parameters are defined initially and collected prospectively. At the end of the prospective study, these data are analyzed.

A big bias is the following: ''These individuals received their cancer care (including chemotherapy and radiation) from numerous different medical providers across the region, but had their surgery completed at the single, high-volume center.'' There could be differences in neoadjuvant therapy in dosage, type of chemotherapy, the period between neoadjuvant therapy and surgery, etc.

Also, two important parameters that influence all complications are serum hemoglobin and albumin levels. This could be different between these two groups and is missing in this study.

Line 169: The age should not be written on 2 decimal points. ''63.78±10.16 years and rural individuals being 169 64±10.46 years of age (P=0.43).''

Line 173: it should be ''radiation'' instead of ''radtion''

Author Response

Reviewer #2:

  1. Line 90: The study is either retrospective or prospective. Even in prospective studies, the results are analyzed ''retrospectively'' because they are analyzed at the end of the prospective period. The main difference between retrospective and prospective studies is that in retrospective studies, the patients went through institutional protocol, and when the idea for the study came out, there was a possibility that there was not all the necessary data in the electronic medical records. In a prospective study, the parameters are defined initially and collected prospectively. At the end of the prospective study, these data are analyzed.

Response: Thank you for the helpful information! We have altered the phrasing in response to Reviewer’s 1 comments also.  We have kept the language of this being a retrospective review of prospectively collected data.  This is an important differentiator compared to studies where the data is collected retrospectively.

  1. A big bias is the following: ''These individuals received their cancer care (including chemotherapy and radiation) from numerous different medical providers across the region, but had their surgery completed at the single, high-volume center.'' There could be differences in neoadjuvant therapy in dosage, type of chemotherapy, the period between neoadjuvant therapy and surgery, etc.

Response: We appreciate the comments and further discussed this as a possible limitation to our study in the discussion section.

  1. Also, two important parameters that influence all complications are serum hemoglobin and albumin levels. This could be different between these two groups and is missing in this study.

Response: We appreciate that you have recognized a very important limitation to our study. Unfortunately, serum hemoglobin and albumin levels were not collected from our data set. We have spent some time discussing this as a limitation in our discussion section and have provided 5 different citations to emphasize the importance. While it is possible that these levels may be statistically different between the two groups and may explain some changes in post-operative complications, it is re-assuring that rates of these complications are overall similar, and men are more likely to get recurrent laryngeal nerve paresis while women were more likely to suffer from post-operative chylothorax. It still may play an unknown role in LOS though.

  1. Line 169: The age should not be written on 2 decimal points. ''63.78±10.16 years and rural individuals being 169 64±10.46 years of age (P=0.43).''

Response: We appreciate the comments. The analysis has been altered to reflect that.

  1. Line 173: it should be ''radiation'' instead of ''radtion''

Response: Thank you for catching that typo. It has been appropriately corrected.

Round 2

Reviewer 2 Report

Comments and Suggestions for Authors

All queries answered.